# Knowledge, attitudes and practices towards malaria in a peri-urban Cameroonian village: Persistent silent transmission despite satisfactory awareness

Steve Joko[1,2]*, Yacouba Poumachu[2], Rehema Ouko[3],
Nelly Armanda Kala-Chouakeu[1,2], Roland Bamou[1,4], Emmanuel Elanga-Ndillé[1,5],
Parfait Awono-Ambene[2], Christophe Antonio-Nkondjio[2], Timoléon Tchuinkam[1],
Cyrille Ndo[5,6]

**1** Department of Animal Biology, Faculty of Science of the University of Dschang, Vector Borne Diseases Laboratory of the Research Unit of Biology and Applied Ecology (VBID-RUBAE), Dschang, Cameroon, **2** Institut de Recherche de Yaoundé, Organisation de Coordination Pour La Lutte Contre Les Endémies en Afrique Centrale (IRY-OCEAC), Yaoundé, Cameroon, **3** Department of Infectious Disease Epidemiology and International Health, London School of Hygiene &Tropical Medicine, London, United Kingdom, **4** Laboratory of Malaria and Vector Research, National Institute of Allergy and Infectious Diseases, National Institutes of health, Rockville, United States of America, **5** Department of Microbiology and Parasitology, Centre for Research in Infectious Diseases (CRID), Yaoundé, Cameroon, **6** Department of Biological Sciences, Faculty of Medicine and Pharmaceutical Sciences, University of Douala, Douala, Cameroon

\* stevejoko0@gmail.com

## Abstract

Malaria remains a major cause of morbidity and mortality in sub-Saharan Africa, including Cameroon, despite ongoing control efforts. Knowledge, Attitudes, and Practices (KAP) studies are essential for identifying behavioural determinants of persistent transmission in communities especially in those of rural areas. To assess knowledge, attitudes, and practices (KAP) towards malaria in a Cameroonian village and investigate their association with parasitological prevalence in a peri-urban Cameroonian village and investigate their association with parasitological prevalence, highlighting silent transmission in the population. A cross-sectional study was conducted in Emana village (Cameroon) involving 249 participants for parasitological screening (microscopy) and a subgroup of 104 with complete sociodemographic data. KAP scores were analysed using non-parametric tests (Mann-Whitney U, Kruskal-Wallis H, Spearman correlation). Malaria prevalence was evaluated using chi-square tests and univariate logistic regression. Categories were regrouped when necessary for complementary analyses. Knowledge and attitudes were generally satisfactory and homogeneous across the population, with no significant association with sociodemographic factors (all p > 0.05). Practices were generally poor but significantly better among women than men (p = 0.014). Overall malaria prevalence was 24.9%, significantly higher in the 5–18 years age group (13.25%, p = 0.027),

**Data availability statement:** All relevant data are within the paper and its Supporting Information files.

**Funding:** The author(s) received no specific funding for this work.

**Competing interests:** The authors have declared that no competing interests exist.

**Abbreviations:** BUCREP, Central Office of Census and Population Studies; INS, Institut Nationale de Statistiques; LITNs, Long-lasting insecticide-treated nets; KAP, Knowledge, attitude, and practice; MINT, Ministry of Transport; NCEP, National Centers for Environmental Prediction; PNLP, Programme National de Lutte contre le Paludisme.

highlighting silent and persistent transmission in this age bracket. No significant associations were observed with occupation or household size; education showed heterogeneous trends depending on category grouping. Despite satisfactory knowledge and positive attitudes, a persistent knowledge-to-practice gap exists, across subgroups, contributing to silent transmission. These findings support targeted interventions: strengthened messaging, and enhanced surveillance in all categories particularly the 5–18 years age group. Complementary qualitative studies could further explore specific practical barriers.

## Introduction

Malaria is one of the most dangerous diseases in the world and remains a real public health problem. Africa is the continent most affected, with around 228 million cases and almost 608,000 deaths. Countries where malaria is endemic are implementing measures such as the use of long-lasting insecticidal nets (LLINs), indoor residual spraying of insecticides (IRS), rapid diagnosis and intermittent preventive treatment of pregnant women and children and recently the use of malaria vaccines for the prevention of *P. falciparum* malaria in children living in malaria endemic areas, to combat this deadly disease [1]. A considerable reduction in the global burden of malaria has been observed since the 2000s [2, 3] but the global burden of malaria remains high despite the control measures undertaken. Cameroon, like many African countries, is highly affected by malaria whit around 6 million cases annually, the disease burden being highest in rural areas [1]. This is due to environmental factors that favor the breeding of mosquitoes, restricted access to health services, and a lack of medical infrastructure.

Asymptomatic malaria characterized by the presence of non-reproductive parasites in the blood, without clinical symptoms or any manifestation of the disease plays a key role in the persistence of transmission, particularly in hyperendemic regions such as the equatorial region of Cameroon [4]. The majority of research describes it as not having a fever or high temperature. However, some other studies consider other criteria such as the absence of fever spells, the lack of fever reported or the absence of malaria treatment before and after the time of the study [5,6]. Individuals infected without showing symptoms may contain *Plasmodium* parasites and act as a reservoir, especially in regions where the infection rate is high. Although they do not show evident signs, these individuals are essential in maintaining the spread of malaria and compromises its elimination [7]. This variant of the disease is often overlooked in control plans because people are not screened or treated, allowing the parasite to persist within the community.

The majority of studies conducted in Cameroon on the prevalence of malaria are based on the diagnosis of populations who do not exhibit any major signs of the disease. In this regard, high prevalences of symptomatic malaria have been reported throughout the country [8,9]. Other studies indicate a link between the attitudes and practices of communities and the levels of prevalence recorded [10,11].

Furthermore, the knowledge, attitudes and practices (KAPs) of rural communities are essential factors in the effective scale-up of malaria prevention and control measures. Studies have shown that behaviour and the perception of risk (KAPs) within communities influence the effectiveness of control strategies such as the use of insecticide-impregnated mosquito nets and early recourse to healthcare [9,12]. Similar studies carried out in other African contexts, such as Nigeria [13] and Uganda [14], have shown that cultural perceptions and the level of awareness of populations have a direct impact on the effectiveness of antimalaria interventions. In rural areas, where means of communication are often limited and traditional practices are common, it is crucial to understand community knowledge and practices regarding the disease in order to improve control interventions.

In this context, the study conducted in Emana, a village in the equatorial region of Cameroon, aims to assess the prevalence of asymptomatic malaria infections and to analyze the KAPs of the inhabitants with regard to the disease. By integrating epidemiological and sociological approaches, this draws on previous work, such as that of [15] in Cameroon, which highlighted the need to combine biological diagnosis with understanding of socio-cultural dynamics in order to design appropriate control strategies. This information is essential for a better understanding of the specific challenges linked to the persistence of malaria and for adjusting control strategies to the local situation.

## Materials and methods

### Study area and target population

Emana is a village situated in the Centre region of Cameroon, in the Lékié division and in the Batchenga subdivision (Fig 1). It is located approximately 40 km from the city of Yaoundé; it lies between latitude 4°16'N and longitude 11°20' E and has a humid savannah tropical climate or savannah climate with dry winters (Aw) according to the Köppen-Geiger

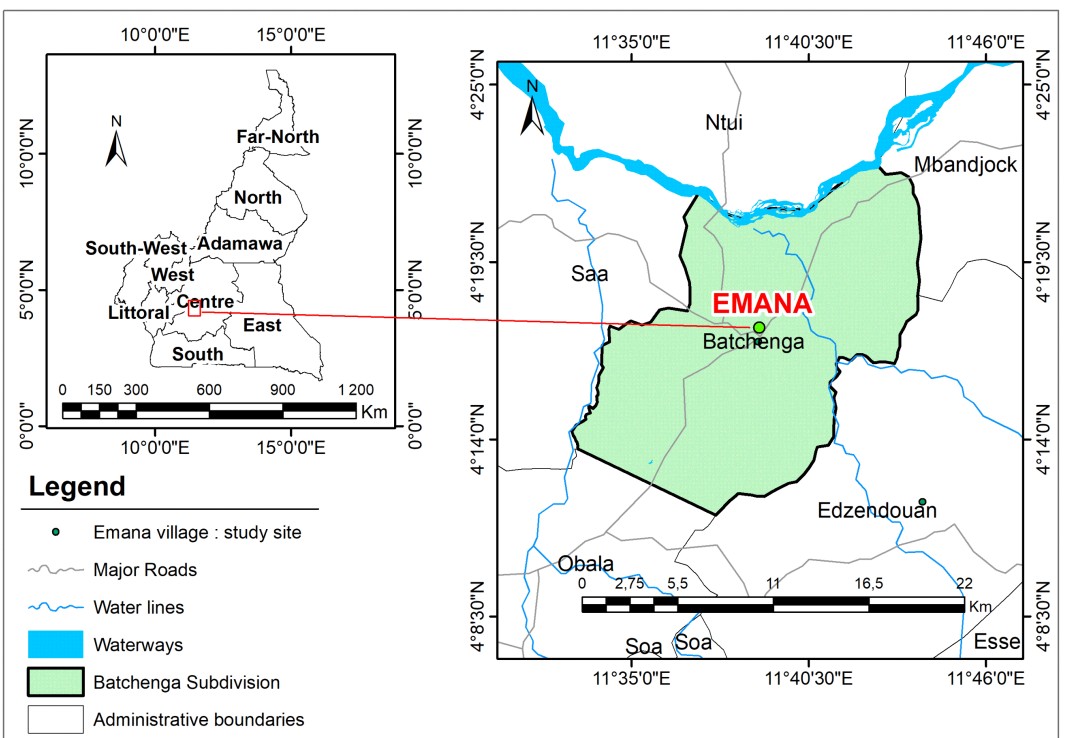

**Fig 1. Study area: The localization village of Emana.**

classification. The locality is characterized by average rainfall ranging from 1200 mm/year to 2000 mm/year on the seacoast, with an annual temperature from 23.5°C to 25 and relative humidity of 83% according to [16]. The climate in the Centre region of Cameroon is made up of two rainy seasons and two dry seasons. It belongs to the southern forest region with a protracted dry season (December-February), followed by a long rainy season (September-November), then a short dry season (July-August) and the short rainy season (March to June) [16]. The vegetation around the village consists of equatorial forest degraded by agricultural activities. The population of Emana is estimated at nearly 600 peoples and agriculture is the major occupation according to data from the chiefdom. The only health center available to the population is located in another locality called Nkolmekok which is situated about 6 km from Emana.

## Study design and population

This cross-sectional study was conducted prospectively in Emana, a rural village in Cameroon to assess people's knowledge, attitudes and practices (KAP) related to malaria prevention and diagnose asymptomatic malaria microscopically using thick smears and blood films. The target population was the residents of Emana who had lived in the village for at least 6 months. Participants including adults and adolescents/children were prospectively recruited from January 28, 2023 to March 27, 2023 through household visits and community engagement. No retrospective data, archived samples, or medical record were utilized in this study. All data collection was performed in real-time by trained fieldworkers and microscopists following WHO guidelines.

## Sample size calculation

Emana is a village with 104 inhabited households and all were sampled for the knowledge, attitudes and practices survey. A representative sample of the population of Emana was taken for malaria diagnosis using the Yamane [17]:

$$\eta = \frac{N}{(1 + N(\varepsilon)2)};$$

$$\eta = \frac{600}{(1 + 600(0,0025))} = 240 \text{ samples for malaria diagnosis}$$

where ɲ is the size of sampling sought; N the population size, and ε the sampling error (5%).

## Inclusion, non-inclusion and exclusion criteria

The samples were collected from individuals of all ages by a qualified nurse. Participants were selected regardless of gender. Only permanent residents of the locality, showing no specific malaria symptoms or fever, were considered. Inclusion in the study required written consent from the participants, along with parental assent for minors. Individuals reporting hemophilia diseases were excluded. In total, 23 individuals were disqualified based on these criteria. Critically ill patients who were unable to answer study questions and those who had taken antimalarial medication in the 4 weeks prior to sample collection were excluded (risk of false negatives or interference with results). Patients clinically suspected of having malaria, were referred to the laboratory of the medical center of Nkolmekok for blood film examination and who gave their consent to participate in the study were not included.

## Collection of blood and parasite detection

Capillary blood was taken from each asymptomatic participant to determine the prevalence of *Plasmodium* in the locality. Thick smears and blood films on the slides were made on site and then transported after blood collection in the same day

to the Organisation de la Coordination pour la Lutte Contre les Endémies en Afrique Centrale (OCEAC) laboratory, where they were stained and read. The reading of the stained slides was carried out by two laboratory technicians to minimise errors. The parasites were identified, and the number of positive cases was recorded.

## Ethical considerations

The study was approved by the National Ethics Committee for Human Health Research of Cameroon under the reference CNERSHC N°001922015. The approval covers the entire study period from January 28, 2023 to March 30, 2023, including the collection of blood samples for thick smears and blood films, and the administration of KAP questionnaires. Written consent was obtained from adult participants and assent for minors was obtained from parents. This was documented and approved by the CNERSHC. Only households for which informed consent of the participants and parental consent for younger children had been obtained were enrolled in the study.

## Statistical analysis of the data

Data were checked for completeness, entered and analyzed using R software version 4.3.3.

**Descriptive statistics.** Knowledge, attitude and practice scores are reported in Table 2. Categorical variables (sex, age groups, occupation, education level, household size categories) are presented as frequencies and percentages. Continuous variables, if any, are summarized as means ± standard deviation (SD) or medians with interquartile ranges (IQR) depending on the distribution (assessed via Shapiro-Wilk test for normality).

For the parasitological data (Table 1), prevalence of malaria (positive slides for trophozoites and/or gametocytes) was calculated as the proportion of positive cases among the total tested individuals (N = 249 overall; N = 104 for sociodemographic subgroups with complete data).

The prevalence of malaria in relation to housing quality was assessed in the full sample (N = 249 for house type) and in the subgroup with complete sociodemographic data (N = 104 for other housing variables). Housing characteristics were treated as categorical variables

**Univariate analyses.** For the KAP components (knowledge, attitudes, and practices scores), which were treated as ordinal or continuous variables (non-normally distributed based on preliminary assessments), differences across sociodemographic categories were evaluated using non-parametric tests:

Comparison between two groups (e.g., sex: male vs female): Mann-Whitney U test. Comparison across more than two groups (e.g., age groups, occupation categories, education levels): Kruskal-Wallis H test. Association with ordinal or

**Table 1. Distribution of malaria cases according to sex, age and type of house in Emana.**

| Variables(N = 249) | | Positive slides to trophozoites N(%) | Positive slides to gametocytes N(%) | p-value |
|---|---|---|---|---|
| **Sex** | Female | 31(24.9%) | 6(2.4%) | 0.13 |
| | Male | 31(24.9%) | 4(1.6%) | |
| **Age groups** | 0-5 | 5(2.01) | 0(0) | 0.02 |
| | 5-18 | 33(13.25) | 5(2.08%) | |
| | 18-96 | 24(09.64) | 2(0.83%) | |
| | Non available | 6(0) | 3(0.125%) | |
| **House type** | Aluminum | 1(0.4%) | 0(0) | 0.5 |
| | Cement | 18(7.23%) | 1(0.4%) | |
| | Mixed | 14(5.62%) | 2(0.8%) | |
| | Mud | 29(11.64%) | 7(2.81%) | |

continuous predictors (e.g., household size, age as ordinal): Spearman's rank correlation coefficient (rho). Results are reported as test statistics (U or H), correlation coefficients (rho), and corresponding p-values.

For the parasitological prevalence data, associations between sociodemographic factors, and also housing factors and malaria positivity (binary outcome: positive vs negative) were assessed using Pearson chi-square test ($\chi^2$) or Fisher's exact test when expected frequencies were <5 in any cell. Crude odds ratios (COR) with 95% confidence intervals (95% CI) were calculated for each category using logistic regression (univariate). Reference categories were chosen as the most frequent or logically baseline group (e.g., females for sex, 5–18 years for age groups, farmers for occupation).

**Bivariate and regrouped analyses (complementary).** When cell counts were low (e.g., for education level or occupation), categories were regrouped to improve statistical power while maintaining biological and contextual relevance: Education: Low level (no education + primary + secondary) vs high level (university). Occupation: Without main occupation (students + unemployed) vs with main occupation. For these regrouped 2 × 2 tables, Fisher's exact test was applied, with odds ratios (OR) and 95% CI reported.

All analyses were univariate/bivariate; no multivariate modeling was performed in this preliminary stage due to the exploratory nature of the study and sample size limitations in certain subgroups. Results are presented in tables with test statistics, p-values, and effect sizes where applicable.

## Results

### Determination of the Prevalence of asymptomatic carrier of malaria in Emana

**Prevalence of asymptomatic carrier of malaria among people examined.** Of the 249 people tested in Emana, the overall prevalence of trophozoites of *Plasmodium falciparum* (active infection) was approximately 25% (62 cases). No significant difference was observed between the sexes (p = 0.13). However, age is significantly associated with positivity (p-value = 0.02), with a marked peak among children and adolescents aged 5–18 years (13.25%). The type of house does not appear to influence the risk of infection (Table 1).

The KAP scores (Table 2) show a moderate-low level of knowledge (mean 5.11 ± 1.35; median 5.0), relatively good attitudes (mean 6.67 ± 2.03; median 7.0) and poor practices (mean 3.54 ± 0.65; median 4.0).

**Assessment of people's knowledge, attitudes and practices towards malaria in Emana. Socio-demographic characteristics of the population studied:** Sociodemographic characteristics of participants are recorded in Table 3. A total of 104 households were surveyed and an average of 6.78 ± 4.25 residents per household were recorded. Each household had at least 1.02 ± 1.32 children under the age of 5. Participants ranged in age from 18 to 83 years, with an average age of 46.94 ± 16.56 years. All were residents of the locality and their main activities were agriculture (45.19%) and commerce (12.5%). Most of the people questioned had received a school education, with half (50%) having a secondary education. The majority of houses were built of earth bricks (53.85%) and breeze-blocks (42.31%). Many of these houses have no eaves (74.04%) or ceilings (96.15%) and have openings in the windows and walls.

**Table 2. Knowledge, attitude and practice scores of the respondents (N = 104).**

|  | Knowledge score (0–12) | Attitude score(0–10) | Practice score(0–8) |
|---|---|---|---|
| **Mean** | 5.11 | 6.67 | 3.54 |
| **Sd** | 1.35 | 2.03 | 0.65 |
| **Median** | 5.0 | 7.0 | 4.0 |
| **IQR** | (4.0, 6.0) | (5.0, 8.0) | (3.0, 4.0) |
| **Min; max** | 1; 9 | 1; 11 | 2; 4 |

**Table 3. Socio-demographic characteristics of participants in the Emana village (N = 104).**

| Characteristics | Categories | Frequency | Percentage |
|---|---|---|---|
| **Sex** | Female | 57 | 54.81 |
| | Male | 47 | 45.19 |
| **Age group** | 18-25 | 9 | 8.64 |
| | 26-35 | 22 | 21.15 |
| | 36-45 | 22 | 21.15 |
| | ≥ 46 | 51 | 49.04 |
| **Mean±SD age** | 46.94 ± 16.56, median = 45.0, IQR = (34.0, 61.0) | | |
| **Household with a number of people** | <5 | 32 | 30.77 |
| | ≥5 | 72 | 69.23 |
| **Mean±SD people length** | 6.79 ± 4.25, median = 6.0, IQR=(4.0,9.0) | | |
| **House type** | Aluminum | 3 | 2.88 |
| | Earth (earth bricks) | 39 | 37.5 |
| | Hard(breeze-blocks) | 44 | 42.31 |
| | Mixed(cement+earth) | 17 | 16.35 |
| | Wood | 1 | 0.96 |
| **Roof type** | Aluminum | 104 | 100 |
| **Eaves** | No | 77 | 74.04 |
| | Yes | 27 | 25.96 |
| **Ceiling** | No | 100 | 96.15 |
| | Yes | 4 | 3.85 |
| **Holes walls** | No | 45 | 43.27 |
| | Yes | 59 | 56.73 |
| **holes windows** | No | 41 | 39.42 |
| | Yes | 63 | 60.58 |
| **holes doors** | No | 73 | 70.19 |
| | Yes | 31 | 29.81 |
| **Occupation** | Farmer | 47 | 45.19 |
| | Trade | 13 | 12.5 |
| | Other occupation | 31 | 29.81 |
| | None | 13 | 12.5 |
| **Education** | University | 10 | 9.62 |
| | Primary | 39 | 37.5 |
| | Secondary | 52 | 50 |
| | No education | 3 | 2.88 |

### Respondents' knowledge about malaria

The study revealed that all respondents had heard of malaria (Table 4). Almost all respondents attributed malaria transmission to the bite of infected mosquitoes (92.3%) while non negligeable proportion (10.57%) attributed it to other causes. Among the major signs and symptoms of malaria, fever (51.9%), nausea/vomiting/loss of appetite (62%), headaches (26.92%) and aches and pains (26.92%) were the most commonly reported. The known methods of malaria prevention were mainly the use of insecticide-treated mosquito nets (90.38%). A very small proportion (4.8%) of respondents indicated the practice of sanitation and hygiene, and the use of insecticides as other malaria prevention methods. With regard

**Table 4. Respondents' knowledge of the signs, symptoms and means of preventing malaria in the Emana village (N = 104).**

| Variables | Categories | Frequencies (%) |
|---|---|---|
| **Do you know what malaria is?** | Yes | 104(100) |
| | No | 0(0) |
| **What causes malaria? *** | The bite of an infected mosquito | 96(92.3) |
| | Other | 11(10.57) |
| | Don't know | 5(4.8) |
| **Major signs and symptoms *** | Fever | 54(51.9) |
| | Headache | 28(26.92) |
| | Aches/Tiredness | 28(26.92) |
| | Chills/Cold | 12(11.53) |
| | Nausea/Vomiting/Lossof appetite/Bitter mouth | 62(59.61) |
| | Other | 21(20.19) |
| **Main transmission season*.** | Rainy season | 34(32.69) |
| | Dry season | 27(25.96) |
| | All year round | 37(35.57) |
| **Methods to prevent malaria*** | LLINs | 94(90.38) |
| | Sanitation | 5(4.8) |
| | Reinforcing the structure of the house | 7(6.7) |
| | Spraying with repellents | 5(4.8) |

*= multiple answers

to the main malaria transmission season, rainy season was the most indicated (32.69%), but a larger number of respondents (35.57%) indicated that malaria was present in both seasons.

### Factors associated with malaria knowledge

None of the socio-demographic factors studied (gender, age, household size, occupation, level of education) are significantly associated with the level of knowledge about malaria at the 5% threshold in univariate analysis. However, larger households (>5 people) tend to have slightly lower levels of knowledge (p-value = 0.064). A higher level of education is associated with a tendency towards better knowledge (H = 6.055, p-value = 0.109) (Table 5).

### Respondent's attitudes toward malaria prevention and control

Of all respondents, 42 (40.3%) reported that they slept under a mosquito net and had slept under a mosquito net the night before the survey. A considerable proportion (42.3%) said that all members of their household slept under a net when present. A small proportion (17.2%) had their children sleep under a net. Looking at the frequency of use of LLINs, a large number of respondents (83.6%) reported using them regularly. However, 12.5% did not use LLINs, and this was mainly attributed to the heat (20.1%). According to the interviewees, an average of 2.83 ± 2.59 peoples under the age of 15 and 3.83 ± 4.03 peoples over the age of 15 had malaria in a year per households. To treat cases of the disease, 45.1% of respondents used medicinal plants, while 43.2% used medicines and 42.3% said they went to hospital to treat malaria cases (Table 6). Forty-six point one (46.1%) and 37.5% of respondents closed their doors and other main openings in their houses between 7 pm to 8 pm and 9 pm to midnight respectively.

**Table 5. Measure of the association between socio-demographic characteristics and the knowledge of the participants in the Emana village.**

| Characteristics | Categories | Applied tests |
|---|---|---|
| **Sex** | Female | U = 1115.50; p-value = 0.1311 |
| | Male | |
| **Age group** | 18-25 | rho=0.068; p-value=0.4916 |
| | 26-35 | |
| | 36-45 | |
| | ≥ 46 | |
| **Number of people** | ≥5 | rho=0.182; p-value=0.0643 |
| | <5 | |
| **Occupation** | Trade | H = 3.048; p-value = 0.3843 |
| | Other | |
| | None | |
| | Agriculture | |
| **Education** | University | H = 6.055; p-value = 0.109 |
| | Primary | |
| | Secondary | |
| | No education | |

*U: Mann–Whitney (M vs F), H: Kruskal–Wallis, rho: Spearman corr*

## Factors associated with malaria attitudes

In univariate analysis, none of the socio-demographic factors studied (gender, age, household size, main occupation, level of education) were significantly associated with attitudes towards malaria (all p-values > 0.05, and well above the threshold of 0.05) (Table 7).

## Respondents' malaria prevention practices

Respondent's practices regarding malaria are recorded in Table 8. Almost all the people questioned (98.07%) said they felt mosquito bites mainly at night, and the main means of protection was the use of an LLIN (90.38%). Very few people reported using insecticides (7.69%) or repellents (1.92%) to protect themselves from mosquito bites. An average of 2,38 ± 1.58 LLINs and 3.81 ± 1.33 bedrooms were recorded. The majority of respondents (68.26%) stated that they had obtained their LLINs from the Ministry of Public Health of Cameroon (MINSANTE); considerable proportions of respondents had acquired their LLINs by purchase (19.23%) and by donations other than from MINSANTE (12.5%). Most respondents (58.65%) said they had held them for more than a year. For malaria cases, the estimated cost of treatment ranged from 1.57$ to 15.27$ for the majority of respondents (44.23%), while a considerable proportion (23.07%) indicated spending between 15.27$ and 76.33$. A very small proportion (6.76%) said they spent nothing on treating cases of malaria, while 20.19% of respondents were unable to estimate the cost of treating cases of the disease.

## Factors associated with malaria practices

Only gender was significantly associated with malaria prevention practices (p = 0.014), with women demonstrating better practices than men (Table 9). No other sociodemographic factors studied (age, household size, occupation, level of education) were significantly related to practices (all p-values > 0.05).

Table 6. Respondents' attitudes towards malaria in the Emana village (N = 104).

| Variables | Categories | Frequencies | Percentages |
|---|---|---|---|
| Who sleeps under a mosquito net and slept there last night | Everyone | 42 | 40.38 |
| | Parents only | 44 | 42.31 |
| | Children aged <5 | 11 | 10.58 |
| | Children aged >5 | 7 | 6.73 |
| How do you use the mosquito net | in the dry season | 0 | 0 |
| | In the rainy season | 4 | 3.85 |
| | Regularly | 6 | 5.77 |
| | Do not use | 94 | 90.38 |
| Reasons not to use a mosquito net regularly when have it | Heat | 21 | 20.19 |
| | By preference | 4 | 3.85 |
| | Regular use | 79 | 75.96 |
| Average number of fevers in one year | People aged <15 | 5.92±6.43 | 1.9 |
| | People aged >15 years | 3.36±2.42 | 2.8 |
| Average of malaria cases in a year | People aged <15 years | 2.83±2.59 | 3.8 |
| | People aged >15 years | 3.83±4.03 | 1.9 |
| How are you treating the disease? * | Going to hospital | 44 | 42.31 |
| | Treating yourself with medicines | 45 | 43.27 |
| | Using herbal remedies | 47 | 45.19 |
| | Combining plants and medicines | 16 | 15.38 |
| Activities at night? | Yes | 31 | 29.81 |
| | No | 73 | 70.19 |
| Closing time for entrances of house | 5pm-6 pm | 13 | 12.5 |
| | 7pm-8 pm | 48 | 46.15 |
| | 9pm-12 pm | 39 | 37.5 |
| | No closing time | 3 | 2.88 |
| | Not specified | 1 | 0.97 |

*Multiple answers

## Association between malaria and sociodemographic factors of people of Emana

Table 10 shows the association between prevalence and the participants' socio-demographic factors. Prevalence was significantly associated with age (p-value = 0.027). People aged between 5 and 18 were the most infected (34.0%) [3.30 (CI:1.26–10.35) p = 0.023] and males were more affected by the disease (29.8%) [1.56 (CI:0.88–2.79), p-value = 0.131]. Malaria was more frequent among people with fewer than five residents (10.0% (CI:0.05–2.12), p-value = 0.29].

The prevalence of malaria in relation to housing quality is shown in the Table 11. No significant association was found between any housing characteristic and malaria prevalence (all p-values > 0.05), although non-significant trends were noted, such as higher susceptibility in houses without ceilings (COR = 0.37, 95% CI: 0.016–8.45, p = 0.612), near stagnant water (COR = 3.07, 95% CI: 0.15–23.63, p = 0.340), or in mud houses (COR = 0.38, 95% CI: 0.01–9.73, p = 0.495). Infinite odds ratios were reported in cases of perfect separation (e.g., zero positives in certain categories).

## Discussion

This study revealed a prevalence of asymptomatic malaria of 24.9% among the population of Emana, a peri-urban area of Yaoundé. This rate remains a cause for concern in view of ongoing control efforts, as it indicates the presence of a

**Table 7. Measure of the association between socio-demographic characteristics and the attitude of the participants in the Emana village.**

| Characteristics | Categories | Applied tests |
|---|---|---|
| **Sex** | Female | U = 1318.00, p-value = 0.8897 |
| | Male | |
| **Age group** | 18-25 | rho=-0.076, p-value=0.4453 |
| | 26-35 | |
| | 36-45 | |
| | ≥ 46 | |
| **Number of people** | ≥5 | rho=-0.086, p-value=0.3831 |
| | <5 | |
| **Occupation** | Trade | H = 1.551, p-value = 0.6706 |
| | Other | |
| | None | |
| | Agriculture | |
| **Education** | University | H = 2.115, p-value = 0.5490 |
| | Primary | |
| | Secondary | |
| | No education | |

*U: Mann–Whitney (M vs F), H: Kruskal–Wallis, rho: Spearman corr*

silent human reservoir of *Plasmodium* parasites, often undetected and untreated, and playing a central role in perpetuating transmission. This result is consistent with observations in other hyperendemic areas of Central Africa such as those reported by [15,18] and [10] who found comparable prevalences in rural Cameroonian communities. Of the positive cases, 4.02% showed gametocytes, the transmissible forms of the parasite to the anopheles. This presence demonstrates that asymptomatic carriers are not only infected but also potentially infectious, and can therefore sustain transmission within the community, even outside clinical peaks [7]. This highlights the urgent need to strengthen active screening strategies and targeted treatment of asymptomatic infections to reduce the parasite reservoir, as recommended in the malaria elimination approaches [1].

Knowledge and attitudes towards malaria were generally satisfactory and homogeneous across sociodemographic groups and no significant associations with age, sex, education, occupation, or household size. This may reflect the effectiveness of national awareness campaigns (community radios, posters, school interventions) that have enabled widespread dissemination of key messages (transmission by mosquitoes, symptoms, prevention). Similar studies in urban and peri-urban areas of sub-Saharan Africa have also reported high and homogeneous levels of knowledge, independent of education level or gender [19–21].

Even more markedly, no significant association was found between attitudes and sociodemographic variables. A relatively good attitude (recognition of severity, acceptance of preventive measures) appears uniformly favourable across subgroups. This finding aligns with levels reported in similar sub-Saharan African and American settings such as the Eastern of Ghana where only 49.4% exhibited positive attitudes [21], and the Northern of Colombia with 55.7% positive attitudes [22]. Our relatively high mean attitude score (6.67/10 ≈ 66.7%) suggests effective dissemination of public health messages in Cameroon, though comparable studies in Senegal with 59% positive among adolescents [23] and Ghana with 48% among pregnant women [24] indicate persistent challenges in translating attitudes into practices across vulnerable

**Table 8. Respondents' practices regarding malaria, means of protection and management of malaria cases (N = 104).**

| Variables | Categories | Frequencies (%) |
|---|---|---|
| **When do you feel the stings** | At night | 102(98.08) |
| | During the day | 2(1.92) |
| **Measures to protect against bites*** | Use of LLINs | 10(9.62) |
| | Use of coils | 2(1.92) |
| | Use of insecticides | 8(7.69) |
| | Use nothing | 94(90.38) |
| **Time of possession of the LLINs** | <6 months | 12(11.54) |
| | >6 months | 8(7.69) |
| | >1year | 61(58.65) |
| **Acquiring LLINs** | Don't know | 22(21.15) |
| | By Purchase | 20(19.23) |
| | By distribution campaigns | 71(68.27) |
| | By donation | 13(12.5) |
| **Average per household (Mean±SD)** | Of LLINs | 2.38 ± 1.58 |
| | Of bedrooms | 3.81 ± 1.33 |
| **LLINs in every room?** | Yes | 36(34.62) |
| | No | 68(65.38) |
| **Does the house act as a barrier?** | Yes | 69(66.35%) |
| | No | 35(33.65%) |
| **Cost of treating malaria cases** | [1.57$-15.27$] | 46(44.23) |
| | [15.27$-76.33$] | 24(23.08) |
| | >76.33$ | 7(6.73) |
| | Spent nothing | 6(5.77) |
| | No idea | 21(20.19) |

*Multiple answers

groups. However, it remains non-effective and this may due to the lack of physical protection (LLINs use of low), outdoor nocturnal activities as mentioned by the respondents and the closure of openings after 6 pm, when the aggressiveness of *Anopheles* mosquitoes begins. These attitudes testify to an underestimation of simple environmental measures such as closing doors and windows and staying indoors in protection against malaria and further expose the population to the risks of infection and perpetuate silent transmission across the population [25].

In contrast to knowledge and attitudes, practices show a significant association only with sex: women adopt better practices than men. This result is consistent with numerous studies in sub-Saharan Africa showing that women, often responsible for child and household health, are more likely to use insecticide-treated nets, seek prompt diagnosis, and adhere to treatments [13,23,26]. The lack of association with education level or occupation suggests that barriers to good practices are not primarily cognitive, but potentially related to gender factors (domestic responsibilities, access to resources)., associated to structural factors (availability, obsolescence of LLINs, perceived effectiveness) or behavioural factors (neglect, heat) indicated by respondents, which merit further exploration. Similar studies in Cameroon have also identified a gap between knowledge about malaria and the effective adoption of preventive practices [27,28]. One of the key findings of the survey was the extensive use of herbal medicine. Thus, 46.1% of heads of household indicated that they treated malaria cases at home using medicinal plants. While this practice is part of a tradition of endogenous care, it

**Table 9. Measure of the association between level of practice and respondents' socio-demographic characteristics.**

| Characteristics | Categories | Applied tests |
|---|---|---|
| **Sex** | Female | U = 1018.50, p-value = 0.0144 |
| | Male | |
| **Age group** | 18-25 | rho=0.095, p-value=0.3355 |
| | 26-35 | |
| | 36-45 | |
| | ≥ 46 | |
| **Number of people** | ≥5 | rho=0.075, p-value=0.4505 |
| | <5 | |
| **Occupation** | trade | H = 2.105, p-value = 0.5508 |
| | Other | |
| | none | |
| | Agriculture | |
| **Education** | University | H = 2.251, p-value = 0.5220 |
| | Primary | |
| | Secondary | |
| | No education | |

*U: Mann–Whitney (M vs F), H: Kruskal–Wallis, rho: Spearman corr*

**Table 10. Prevalence of malaria in relation to socio-demographic factors.**

| Characteristics | Category | Frequencies | Prevalence | p-value | Univariate COR (95% CI), p-value |
|---|---|---|---|---|---|
| **Gender (N=249)** | F<br>M | 145(58.23)<br>104(41.77) | 31(21.4)<br>31(29.8)<br>OV: 24.9%<br>$\chi^2$=1.872 | 0.171 | 1<br>1.56 (0.88-2.79),0.131 |
| **Ages groups (n=249)** | 1-5<br>5-18<br>18-96<br>Mean: 24.18<br>SD: 21.56 | 37<br>97<br>109<br>N.A: 6 | 5(13.5)<br>33(34.0)<br>24(22.0)<br>$\chi^2$=7.198 | 0.027* | 0,30, p-value 0.019<br>1,83, p-value = 0.062 |
| **Occupation (N=104)** | Farmer<br>Trader<br>Student<br>Household<br>Others<br>Unemployed | 47<br>13<br>5<br>9<br>17<br>12 | 2 (4.25)<br>1 (7.69)<br>0<br>1 (0.11)<br>2 (11.76)<br>1 (8.33)<br>$\chi^2$= 1.82 | 0.872 | 1<br>1.88(0.157-22.5)<br>2.05(0.082-51.2)<br>2.81(0.227-34.7)<br>3(0.388-23.2)<br>22.52(1.56-325) |
| **Education level (n=104)** | Lower(No educa-tion/Secondary/Primary)<br>Higher(University) | 91<br>13 | 7 (7.7%)<br>13 (100)<br>$\chi^2$=72.5 | 0.0001 | – |
| **Number of people (N=104)** | <5<br>>5 | 50<br>54 | 5 (10.0)<br>2 (3.7)<br>$\chi^2$=0.7899 | 0.374 | 0.39 (0.05-2.12), 0.29 |

*Significant statistical difference observed, COR: Crude ODD Ratio analysis; N.A: Non-Available; $\chi^2$: Chi square test, OV: overall.

**Table 11. Prevalence of malaria in association with housing characteristics.**

| Characteristics (N=104) | Category | Frequencies | Prevalence | P-value | Univariate COR (95% CI),p-value |
|---|---|---|---|---|---|
| House type (N=249) | Aluminum<br>mud<br>Hard<br>Mixed (cement+mud) | 2<br>106<br>86<br>55 | 1<br>29(27.4)<br>18(20.9)<br>14(25.5)<br>χ²=1.7502 | 0.6117 | 1<br>0.38 (0.01-9.73),0.495<br>0.26 (0.01-6.91), 0.356<br>0.34 (0.01-9.02), 0.458 |
| Presence of ceiling? | No<br>Yes | 100<br>4 | 7(7.0)<br>0<br>F=0.5837 | 0,612 | 0,37(0,016−8,45) |
| Presence of eave? | No<br>Yes | 77<br>27 | 1(3.1)<br>6(7.8)<br>χ²=0.5323 | 0.6722 | 1<br>0.46 (0.02-2.84), 0.476 |
| Windows not properly closed or no windows | No<br>Yes | 41<br>63 | 2(4.9)<br>7(7.9)<br>χ²=0.3701 | 0.6952 | 1<br>1.68 (0.34-12.15), 0.547 |
| Doors not properly closed or fitted | No<br>Yes | 73<br>31 | 4(5.5)<br>3(9.7)<br>χ²=0.6108 | 0.6108 | 1.85 (0.35-8.91), 0.440 |
| Cracked or perforated walls? | No<br>Yes | 45<br>59 | 2(4.4)<br>5(8.5)<br>χ²=0.6605 | 0.4628 | 1<br>1.99 (0.41-14.38), 0.424 |
| Number of bedrooms | 1,2<br>2,4<br>4,8 | 6<br>41<br>11 | 2(33.3)<br>4(9.8)<br>0<br>χ²=4.7034 | 0.0980 | – |
| Presence of vegetation in proximity | No<br>Yes | 0<br>104 | 0<br>7(6.7)<br>χ²=77.885 | – | – |
| Use of Windows net | No<br>Yes | 102<br>2 | 7(6.9)<br>0<br>F=0.1471 | 0,52 | 13,85(0,25-750) |
| Stagnant water in proximity | No<br>Yes | 98<br>6 | 6(6.1)<br>1(16.7)<br>χ²=1.003 | 0.3358 | 3.07 (0.15-23.63) 0.340 |
| Use of LLINs | No<br>Yes | 94<br>10 | 7(7.4)<br>0<br>χ²=.0.7984 | – | – |

*Significant statistical difference observed, χ²: Pearson's Chi square test, F: Fisher test, COR: Crude ODD Ratio analysis; -: Infinite values for odds ratios (perfect separation)*

raises questions about therapeutic efficacy, the risk of delayed treatment and the potential for the development of resistance when conventional treatments are avoided, interrupted or taken only as a last resort in the event of complications. This rate is relatively high compared with that of the [11] and [9] studies, where 8.6% and 14.6% of respondents respectively had recourse to herbal medicine treatment for malaria.

## Strengths and Limitation

Strengths include the community-based integration of parasitological and KAP data in a peri-urban setting, providing holistic insights into silent transmission [29]. Non-parametric analyses preserved KAP score integrity, aligning with similar sub-Saharan African studies [30,31].

Limitations encompass the modest sample size (N = 104 for sociodemographics), reducing power for subtle associations. Multivariate modeling was infeasible due to sparse positives (~20 events) and non-significant univariates, risking overfitting common in exploratory KAP research [8,9,28]. Self-reported data may introduce bias, though triangulation mitigates this. Compared to literature, our 24.9% prevalence matches Cameroonian rates (25–66.9%) [11,32], but homogeneous knowledge contrasts with variable levels in Malaysia (46% transmission awareness) [30], suggesting campaign success but persistent practice barriers.

### Implications

For clinicians, findings advocate routine screening to detect silent carriers, reducing reservoirs [1]. Policymakers should address the practice gap through gender-targeted campaigns (e.g., male-focused, LLIN promotion) and environmental interventions (e.g., subsidized housing modifications), integrating into Cameroon's Malaria Program [31]. These could lower prevalence, as seen in sub-sahara Africa where improved adherence reduced infections by 20–40% [29].

### Future Directions

Unaddressed items (e.g., perceived risk, stigma, LLIN costs, family decisions, myths) warrant qualitative studies to unpack barriers. Larger cohorts with multivariable analyses could identify confounders, enhancing elimination strategies [33].

### Conclusion

In summary, despite positive attitudes and basic knowledge, preventive behaviors remain limited with better awareness among women and contributes to ongoing silent transmission in Emana. This study highlights the significant burden of asymptomatic malaria infections in Emana and the importance of KAPs in the fight against the disease. By building on women's knowledge strengths and increasing overall awareness, interventions in Emana could not only reduce local transmission, but also serve as a model for other rural communities in Cameroon facing similar challenges.

### Supporting information

**S1 Dataset. (249)Malaria_Prevalence_Emana_2023_Update.**
(XLSX)

**S2 Dataset. (104)HOH_Malaria_Prevalence_Emana_2023_Update.**
(XLSX)

**S3 Dataset. KAP_Scores_Emana_2023.**
(XLSX)

**S4 Dataset. Sociodemographics and KAP datas_Emana_2023.**
(XLSX)

### Acknowledgments

The authors are grateful to the people of Emana village for their participation in this study, and to the research unit of OCEAC where the slides were observed for malaria parasite diagnostic.

### Author contributions

**Conceptualization:** Steve Joko, Roland Bamou, Timoléon Tchuinkam, Cyrille Ndo.

**Data curation:** Steve Joko, Rehema Ouko.

**Formal analysis:** Steve Joko, Rehema Ouko.

**Funding acquisition:** Steve Joko.

**Investigation:** Steve Joko, Yacouba Poumachu, Nelly Armanda Kala-Chouakeu.

**Methodology:** Steve Joko, Roland Bamou, Emmanuel Elanga-Ndillé, Parfait Awono-Ambene, Christophe Antonio-Nkondjio, Timoléon Tchuinkam, Cyrille Ndo.

**Resources:** Parfait Awono-Ambene, Christophe Antonio-Nkondjio.

**Software:** Parfait Awono-Ambene.

**Supervision:** Nelly Armanda Kala-Chouakeu, Emmanuel Elanga-Ndillé.

**Validation:** Cyrille Ndo.

**Visualization:** Steve Joko, Rehema Ouko, Cyrille Ndo.

**Writing – original draft:** Steve Joko.

**Writing – review & editing:** Steve Joko, Yacouba Poumachu, Rehema Ouko, Nelly Armanda Kala-Chouakeu, Roland Bamou, Emmanuel Elanga-Ndillé, Parfait Awono-Ambene, Christophe Antonio-Nkondjio, Timoléon Tchuinkam, Cyrille Ndo.

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
