## [Decision Letter · Decision Letter 0]

11 Feb 2026

PONE-D-25-68941Knowledge, attitudes and practices Towards Malaria in Cameroonian Village: Low Awareness and Silent Transmission Among YouthPLOS One

Dear Dr. Joko,

Thank you for submitting your manuscript to PLOS ONE. After careful consideration, we feel that it has merit but does not fully meet PLOS ONE’s publication criteria as it currently stands. Therefore, we invite you to submit a revised version of the manuscript that addresses the points raised during the review process.

We look forward to receiving your revised manuscript.

Kind regards,

Clement Ameh Yaro, Ph.D

Academic Editor

PLOS One

Journal Requirements:

Reviewers' comments:

Reviewer's Responses to Questions

**Comments to the Author**

1. Is the manuscript technically sound, and do the data support the conclusions?

Reviewer #1: Yes

Reviewer #2: Yes

2. Has the statistical analysis been performed appropriately and rigorously? 

Reviewer #1: I Don't Know

Reviewer #2: Yes

3. Have the authors made all data underlying the findings in their manuscript fully available?

Reviewer #1: Yes

Reviewer #2: No

4. Is the manuscript presented in an intelligible fashion and written in standard English?

Reviewer #1: Yes

Reviewer #2: Yes

5. Review Comments to the Author

Reviewer #1: In this paper, the authors present the original results of a study that assessed the knowledge, attitudes and practices of a sample of the Cameroonian population regarding silent malaria transmission. The study is well designed, and the results are interesting.

However, we have several suggestions for improving the submitted manuscript.

The text requires complete revision due to numerous spelling and grammatical errors.

The title of the study focuses on young people. To which age group are they referring? Children and adolescents? Or young adults? This point needs to be clarified in the study. The socio-demographic data presented here refer to subjects over the age of 18, whereas the biological results presented refer to children. We have no sociodemographic data on children under 18!

The absence of multivariate analysis is a significant limitation of this study.

The discussion is rather disjointed and seems to have strayed from the initial objectives of the study. This causes the study to lose its original substance. The discussion must be completely rewritten according to the following plan:

- Presentation of the main results, with a focus on the primary and secondary objectives of the study.

- Strengths and weaknesses (limitations) of the study, comparing the results obtained with those in the literature. Only the most significant results should be discussed.

- Implications of the results for clinicians and policymakers

- Questions not addressed in this study and future research in the field

- The references should be reviewed, particularly numbers 1, 3, 10, 13, 17, 21, 23 and 28.

Introduction

Review the English, paying particular attention to the terms 'key (not kay) role' (page 6, line 63) and 'national triangle' (page 9, line 75), which should be replaced by 'Cameroon'.

Methods

Review the English, paying particular attention to the following terms: March (page 10, line 107) and Knowledge (page 12, line 159).

The authors should explain why they chose Emana village for this study. This choice may impact the generalisability of the results.

Why wait four years to publish the results of a study conducted in 2022? (Page 11, line 119).

On page 11, line 134, what do you mean by 'hemolytic diseases'? How could participants with limited medical knowledge tell the difference between this term and other medical conditions?

Results: Page 15, line 225, Table 3: Specify clearly what the term 'number of people' refers to? Participants?

The abbreviation 'LLINs' appears without having been defined earlier, notably on page 17, line 239, in Table 4 and on pages 18 and 19, lines 258 and 259. Does it refer to mosquito nets? If so, please ensure that the abbreviations are consistent (see page 8, line 53).

Review the presentation of results in Tables 5, 7, 9, 10 and 11. I suggest keeping only the p-Value and deleting all intermediate values obtained from the various statistical tests (U: Mann–Whitney (M vs F), H: Kruskal–Wallis, rho: Spearman corr).

p-Value: Ensure that all terms referring to the p-value (p, P-value, …) are harmonised.

You have chosen to display the values in US dollars ($). Remove the term FCFA from page 20, lines 286 to 288, and page 21, line 291, Table 8.

Reviewer #2: Abstract:

The abstract is generally clear and well-written, providing a good overview of the study’s objective, methods, results, and conclusion. However, the overall malaria prevalence of 24.9% is smaller compared to that of a particular age group (5–18 years), which is 34%.

Way Forward

Recalculate the overall prevalence. Taking age, for instance, the overall prevalence should be the sum of the prevalence for each age group.

Introduction/Background:

The introduction is well-written, capturing a strong statement of the problem and outlining clear objectives of the study.

Results:

There are inconsistencies in the results. There are cases of fractional parts of percentage values being mixed up, and commas used in place of points across all the Tables. There is also a mix-up in the use of sample size and number of households.

Way Forward

1. Authors are required to correct all as appropriate. For example, in Table 1, the row for sex, Female 31(24,9%), is corrected to 31(24.9%). All analyses should be done with respect to sample size, not the number of households.

2. The focus is on salient transmission among youth as reflected in the title, but participants of all ages are included in the study. The authors may need to make the necessary adjustments.

3. In most Tables, especially Table 6, their frequency counts are not consistent and are not equal to the sample size.

4. Lines 207-208, the mean values presented should not be further divided by different scores so as to tally with the mean values in Table 2.

5. The rationales for the adoption of different statistical tools should be provided.

Limitations:

Discuss the limitations, including potential sampling biases and issues related to the generalizability to larger populations.

6. PLOS authors have the option to publish the peer review history of their article (what does this mean?). If published, this will include your full peer review and any attached files.

Reviewer #1: **Yes:** Fernando KEMTA LEKPA

Reviewer #2: **Yes:** Emmanuel Alphonsus Akpan

---

## [Author Response · Author response to Decision Letter 1]

8 Mar 2026

Editor comment: We not that your Data Availability Statement is currently as follows: All relevant data are within the manuscript and it’s supporting information files. {…} Author must share their “minimal dataset” for their submission required to replicate the results of your study.

Response: We shared the K.A.P datasets and the parasitological dataset in “other files”

Reviewer 1 comment: The text requires complete revision due to numerous spelling and grammatical errors.

Response: The text was entirely checked and spelling and grammatical errors were corrected.

---

## [Decision Letter · Decision Letter 1]

20 Apr 2026

Knowledge, attitudes and practices towards malaria in a peri-urban Cameroonian village: Persistent silent transmission despite satisfactory awareness

PONE-D-25-68941R1

Dear Dr. Joko,

We’re pleased to inform you that your manuscript has been judged scientifically suitable for publication and will be formally accepted for publication once it meets all outstanding technical requirements.

Kind regards,

Clement Ameh Yaro, Ph.D

Academic Editor

PLOS One

Additional Editor Comments (optional):

Reviewers' comments:

Reviewer's Responses to Questions

**Comments to the Author**

1. If the authors have adequately addressed your comments raised in a previous round of review and you feel that this manuscript is now acceptable for publication, you may indicate that here to bypass the “Comments to the Author” section, enter your conflict of interest statement in the “Confidential to Editor” section, and submit your "Accept" recommendation.

Reviewer #1: All comments have been addressed

Reviewer #2: All comments have been addressed

2. Is the manuscript technically sound, and do the data support the conclusions?

Reviewer #1: Yes

Reviewer #2: Yes

3. Has the statistical analysis been performed appropriately and rigorously? 

Reviewer #1: Yes

Reviewer #2: Yes

4. Have the authors made all data underlying the findings in their manuscript fully available?

Reviewer #1: Yes

Reviewer #2: Yes

5. Is the manuscript presented in an intelligible fashion and written in standard English?

Reviewer #1: Yes

Reviewer #2: Yes

6. Review Comments to the Author

Reviewer #1: I have carefully reviewed the revised manuscript.

I believe the authors have made the requested changes and provided clear explanations to the questions raised.

Reviewer #2: A RECOMMENDATION ON KNOWLEDGE, ATTITUDES AND PRACTICES TOWARDS MALARIA IN PERI-URBAN CAMEROONIAN VILLAGE: PERSISTENT SILENT TRANSMISSION DESPITE SATISFACTORY AWARENESS BY DR. EMMANUEL AKPAN

Comments

The authors have responded adequately to all my review comments, and the manuscript can be accepted for publication.

Abstract:

The abstract is generally clear and well-written, providing a good overview of the study’s objective, methods, results, and conclusion. However, the overall malaria prevalence of 24.9% is smaller compared to that of a particular age group (5–18 years), which is 34%.

Way Forward

Recalculate the overall prevalence. Taking age, for instance, the overall prevalence should be the sum of the prevalence for each age group.

Response

We thank the reviewer for this observation. The overall prevalence of 24.9% is correctly calculated as the proportion of positive slides among the total sample tested (62 positives / 249 participants). The prevalence in the 5-18 years group was recalculated (13.25%). We have revised this in the revised manuscript (Tableau 1, lines 205-206).

The malaria prevalence for the 5-18-year age group has been recalculated from 34% to 13.25%, which is now correct as required. I am satisfied with the response provided.

Introduction/Background:

The introduction is well-written, capturing a strong statement of the problem and outlining clear objectives of the study.

Results:

There are inconsistencies in the results. There are cases of fractional parts of percentage values being mixed up, and commas used in place of points across all the Tables. There is also a mix-up in the use of sample size and number of households.

Way Forward

1. Authors are required to correct all as appropriate. For example, in Table 1, the row for sex, Female 31(24,9%), is corrected to 31(24.9%). All analyses should be done with respect to sample size, not the number of households.

Response

We apologize for these formatting inconsistencies and have corrected them throughout the manuscript in all tables. All decimal separators are now points (e.g., 24.9% instead of 24,9%). Percentages are consistently rounded conveniently (e.g., 24.9%, 34.0%). All analyses and frequencies are now explicitly based on the correct sample size (N=249 for parasitological data; N=104 for sociodemographic and housing variables with complete data). References to "number of households" have been corrected to "number of individual" in a household in Table 11. These changes are visible in the Track Changes version (line 307-308).

All required corrections are effected with a great level of satisfaction

2. The focus is on salient transmission among youth as reflected in the title, but participants of all ages are included in the study. The authors may need to make the necessary adjustments.

Response

appreciate this feedback and agree that the title should accurately reflect the study's scope. The study included participants of all ages for a comprehensive population assessment. To align the title with the content and avoid misleading emphasis on youth alone, we have revised the title to: " Knowledge, attitudes and practices towards malaria in a peri-urban Cameroonian village: persistent silent transmission despite satisfactory awareness" This better reflects the general population focus while retaining the key finding. The revised title appears in the manuscript header and abstract (line 1). We have also clarified throughout the text that the study population includes all ages, with age-stratified prevalence highlighted as a subgroup finding.

The title has been rephrased to capture the scope of the study. I totally agree with the amended title.

3. In most Tables, especially Table 6, their frequency counts are not consistent and are not equal to the sample size.

Response

We have carefully rechecked and corrected all tables for consistency. Frequency counts now sum exactly to the stated sample size (N=249 for parasitological tables; N=104 for sociodemographic/housing tables). Missing or incomplete data are now explicitly noted as "NA". For example, Table 6 has been revised to ensure totals match , except for the variables which have the “*” asterisk indicating a multiple choices or responses. All changes are tracked (Tables 1–10 revised).

I am satisfied with the corrections made.

4. Lines 207-208, the mean values presented should not be further divided by different scores so as to tally with the mean values in Table 2.

Response

This was an editorial error. We have removed any inappropriate division of means by sub scores and ensured consistency with Table 2. The revised text (lines 211–217) reads: "Knowledge scores averaged 5.11 (SD 1.35), attitudes 6.67 (SD 2.03), and practices 3.54 (SD 0.65), as detailed in Table 2."

Corrected as required with a good satisfaction.

5. The rationales for the adoption of different statistical tools should be provided.

Response

We have added a dedicated paragraph in the Methods section (Statistical analysis subsection, page 8, lines 158-196) explaining the choice of tests: "Due to the non-normal distribution of KAP scores (Shapiro-Wilk test), non-parametric tests were used: Mann-Whitney U for two-group comparisons (e.g., sex), Kruskal-Wallis H for multiple groups (e.g., age, occupation, education), and Spearman's rho for correlations (e.g., household size). For binary parasitological prevalence, Pearson's chi-square or Fisher's exact test was applied when appropriate, with univariate logistic regression for crude odds ratios (COR) and 95% CI. These choices preserved data integrity and avoided assumptions violated by parametric tests." This rationale is explicit and justified.

I am satisfied with the justification provided.

Limitations:

Discuss the limitations, including potential sampling biases and issues related to the generalizability to larger populations.

Response

The discussion was rewritten accordingly to the proposed plan by Reviewer 1. A new subsections "Strengths and Limitations", “Implication for clinicians and policymakers” and “future research” were been added to the Discussion (pages 20, lines 320-378): "Strengths include the community-based integration of parasitological and KAP data in a peri-urban setting. Limitations encompass the modest sample size (N=104 for sociodemographics), limiting power for subtle associations. Multivariate modeling was not performed due to sparse positives and non-significant univariates. Self-reported KAP may introduce social desirability bias, though triangulation with prevalence mitigates this. Findings are specific to Emana village and may not generalize to larger or more rural populations due to peri-urban context and potential sampling biases (convenience sampling)." This addresses generalizability and biases directly.

The discussion of limitations has been provided and is satisfactory.

7. PLOS authors have the option to publish the peer review history of their article (what does this mean?). If published, this will include your full peer review and any attached files.

Reviewer #1: **Yes:** Fernando Kemta Lekpa

Reviewer #2: **Yes:** Emmanuel Alphonsus Akpan

---

## [Editor Report · Acceptance letter]

PONE-D-25-68941R1

PLOS One

Dear Dr. Joko,

I'm pleased to inform you that your manuscript has been deemed suitable for publication in PLOS One. Congratulations! Your manuscript is now being handed over to our production team.

Kind regards,

on behalf of

Dr. Clement Ameh Yaro

Academic Editor

PLOS One